# Long Non-Coding RNAs as Novel Targets for Phytochemicals to Cease Cancer Metastasis

**DOI:** 10.3390/molecules28030987

**Published:** 2023-01-18

**Authors:** Sadegh Rajabi, Huda Fatima Rajani, Niloufar Mohammadkhani, Andrés Alexis Ramírez-Coronel, Mahsa Maleki, Marc Maresca, Homa Hajimehdipoor

**Affiliations:** 1Traditional Medicine and Materia Medica Research Center, Shahid Beheshti University of Medical Sciences, Tehran 1434875451, Iran; 2Department of Immunology, Max Rady College of Medicine, University of Manitoba, Winnipeg, MB R3E0T5, Canada; 3Department of Clinical Biochemistry, School of Medicine, Shahid Beheshti University of Medical Sciences, Tehran 1985717443, Iran; 4Research Group in Educational Statistics, National University of Education (UNAE), Azogues 030101, Ecuador; 5Azogues Campus Nursing Career, Health and Behavior Research Group (HBR), Psychometry and Ethology Laboratory, Catholic University of Cuenca, Cuenca 010107, Ecuador; 6Psychology Department, University of Palermo, Buenos Aires 1175ABT, Argentina; 7Epidemiology and Biostatistics Research Group, CES University, Medellín 050022, Colombia; 8Department of Molecular Genetics, Faculty of Biological Sciences, Tarbiat Modares University, Tehran 331-14115, Iran; 9Aix Marseille University, CNRS, Centrale Marseille, iSm2, 13397 Marseille, France; 10Traditional Medicine and Materia Medica Research Center and Department of Traditional Pharmacy, School of Traditional Medicine, Shahid Beheshti University of Medical Sciences, Tehran 1516745811, Iran

**Keywords:** long non-coding RNA, LncRNAs, phytochemical, cancer, metastasis, therapy

## Abstract

Metastasis is a multi-step phenomenon during cancer development leading to the propagation of cancer cells to distant organ(s). According to estimations, metastasis results in over 90% of cancer-associated death around the globe. Long non-coding RNAs (LncRNAs) are a group of regulatory RNA molecules more than 200 base pairs in length. The main regulatory activity of these molecules is the modulation of gene expression. They have been reported to affect different stages of cancer development including proliferation, apoptosis, migration, invasion, and metastasis. An increasing number of medical data reports indicate the probable function of LncRNAs in the metastatic spread of different cancers. Phytochemical compounds, as the bioactive agents of plants, show several health benefits with a variety of biological activities. Several phytochemicals have been demonstrated to target LncRNAs to defeat cancer. This review article briefly describes the metastasis steps, summarizes data on some well-established LncRNAs with a role in metastasis, and identifies the phytochemicals with an ability to suppress cancer metastasis by targeting LncRNAs.

## 1. Introduction

Cancer metastasis is a multi-step process that results in the spread of cancer cells to distant tissues and organs beyond the primary site [1]. This phenomenon is responsible for more than 90% of cancer-related mortality in the world [2]. According to estimations, about 50% of cancer patients already have clinically detectable metastases at the time of initial diagnosis. However, metastasis is most often due to the recurrence of the disease mainly after definitive treatment [3].

Long non-coding RNAs (LncRNAs) are members of the non-coding RNAs family that are more than 200 base pairs in length. These molecules are transcribed from various regions of the genome such as introns, exons, and intergenic connections. To date, nearly 30,000 LncRNAs have been recognized in humans and mice, but the function of only some of them has been recognized [4]. The main role of these RNA molecules is the regulation of gene expression by acting on a variety of intracellular processes from transcription to translation and interfering with signaling pathways [5]. As LncRNAs regulate gene expression, they have critical effects on the proliferation, apoptosis, migration, invasion, and metastasis of cancer cells [6]. A growing body of evidence suggests the potential role of LncRNAs in different steps tumor metastasis, which is the critical stage in cancer progression leading to decreased patient survival [7]. Therefore, targeting these novel and important molecules in cancers may enable the development of treatment methods for this disease.

Phytochemicals are natural bioactive ingredients of a variety of plants with beneficial health effects beyond basic nutrition [8]. They exhibit a number of desirable biological activities including anti-cancer, anti-inflammatory, anti-oxidant, and antimicrobial effects in vitro and in vivo [9]. Phytochemicals exert anti-cancer effects through different mechanisms. They induce cell death in cancer cells, target specific molecules in cellular pathways, modulate oxidative stress, and prevent tumors angiogenesis, which hinders metastasis [10]. A variety of phytochemicals have been shown to inhibit the metastatic propagation of cancer cells via several mechanisms. For example, curcumin is a polyphenol derived from *Curcuma longa* with ability to hamper metastasis of cancer cells by inhibiting transcription factors, cell adhesion molecules, cell surface markers, and epithelial-mesenchymal transition (EMT) [11]. As before mentioned, LncRNAs have a key role in the metastatic spread of cancer cells [7]. Therefore, targeting these regulatory RNA molecules by phytochemicals is of great importance in the treatment of cancer. This review paper summarizes the metastasis process and the well-established LncRNAs involved in this process. It particularly provides a list of phytochemicals that have been used to target LncRNAs for cancer therapy.

## 2. Metastasis and Its Compartments

Metastasis formation involves different steps that are summarized in Figure 1.

### 2.1. Invasion

The invasion of tumor cells occurs due to the loss of attachment and polarity, modification of extracellular matrix (ECM), and alteration of migratory characteristics in these cells. Before metastasis, tumor cells are located in the luminal space, which is different from the stromal compartment [12]. The metastatic tumor cells undergo the EMT process to be ready for invasion and detachment from the primary tumor. This process is achieved by activating a number of cell signaling pathways that are triggered by the transforming growth factor (TGF-β), Wnt ligand, and tyrosine kinase receptors. Genetic alterations also support the activation of EMT-related transcription factors involved in the modulation of cell adhesion and polarity. The lncRNAs affect the expression of these transcription factors [13]. Post-transcriptional modifications and splicing also facilitate this process [14,15]. For example, loss of expression or downregulation of E-cadherin, a cell-cell adhesion protein, is associated with the recruitment and accumulation of circulating tumor cells (CTC) and their intravasation into the blood vessels [16]. Expression of E-cadherin is suppressed by the action of EMT-related factors such as Snail, Slug, Zeb1, and Zeb2, which bind to the promoter region of E-cadherin and lead to the formation of repressive chromatin structure [17]. Mesenchymal features of tumor cells are characterized by a bipolar structure with higher mobility, stemness, and invasiveness. Vimentin, epithelial cytokeratin 8, LC3B, alanine aminopeptidase, occludin, fibronectin, mastermind-like protein 1, myocardin-related transcription factors, some LncRNAs, and various signaling molecules are known to contribute to the EMT [18,19,20,21,22,23,24].

### 2.2. Angiogenesis

Tumor angiogenesis differs from normal angiogenesis in terms of endothelial cell mitogens and chemo-attractants. Tumor neovascularization is characterized by the invasion of the basement membrane toward the primary blood vessel resulting in vessel growth. With the growth of the tumor, primary tumor cells go farther from the blood vessels, the process that leads to hypoxia in the tumor tissue [25]. Hypoxia instigates the production of factors such as vascular endothelial growth factor (VEGF) and fibroblast growth factor (FGF), which promote angiogenesis. Production of some enzymes causes the degradation of the basement membrane of the capillary followed by migration and subsequent proliferation of epithelial cells. Overproduction of VEGF makes tumor vessels leaky and highly permeable leading to increased fluidity in the tumor microenvironment and interstitial pressure [26].

### 2.3. Intravasation

Intravasation is a crucial step in tumor propagation in which tumor cells penetrate vessel walls and enter circulation. This process leads to the dissemination of CTCs into circulation and their movement toward the metastatic site. The entry of tumor cells into lymph vessels is relatively easier than blood vessels because lymph vessels are devoid of the endothelial junction [27]. Through intravasation, tumor cells break through a dense ECM to enter the vessel [28]. Amoeboid intravasation is facilitated by the Rho/ROCK signaling pathway that leads to the formation of blebs. The production of VEGF increases the permeability of endothelial cells for tumor cells [29]. Furthermore, the tumor microenvironment of metastasis plays an important role in the recruitment of tumor cells through chemotactic signals. It has been reported that the presence of CD68^+^ macrophages and CD31^+^ endothelial cells in the vicinity of breast cancer cells instigates hematogenous metastasis [30,31].

### 2.4. Tumor Cell in Circulation

Due to the loss of ECM adhesion, tumor cells encounter a great amount of stress in circulation. Tumor cells need to survive in circulation in order to extravasate and disseminate in a distant organ. These tumor cells form aggregates with blood cells in the circulation; however, in smaller vessels like capillaries, aggregates are modified into chains to allow the passage of the cluster. This helps tumor cells to overcome mechanical shear stress in the bloodstream [32,33]. The expression of mutated pannexin-1, a membrane channel, supports mechanical stress and inhibits the apoptosis of CTCs [34]. 

Loss of adhesion to the ECM, in an integrin-dependent way, can trigger anoikis, which is a programmed cell death induced by the detachment of metastatic cancer cell from the ECM. The tumor cells have to develop anoikis resistance to survive in circulation [35]. To escape anoikis, the tumor cells tend to activate several cell signaling pathways by a zinc-finger transcription factor, FAK, phosphatase and tensin homolog, tyrosine kinases, insulin-like growth factor, and PI3K/Akt [36,37]. Furthermore, these cells escape from immune cells in circulation by the secretion of immunoregulatory molecules that protects them from natural killer cells [38]. Vascular cell adhesion molecule 1 (VCAM1) and vascular adhesion protein 1 (VAP1) also recruit macrophages by expressing tissue factor, which results in blood clotting and facilitates tumor cell survival [39].

### 2.5. Extravasation

Following successful survival in circulation, tumor cells extravasate to a secondary organ. The process involves the adhesion of these cells to the endothelium of blood vessels, alteration in the endothelial barrier to cross it, and migration into underlying tissue. This can or cannot be preceded by the proliferation and differentiation of cancer cells in the blood vessels. Disruption of the endothelial cell-to-cell barrier is an important step in this process [40]. Tumor cells adhere to endothelium by producing several cell adhesion molecules like cadherins, selectins, and integrins [41,42,43]. Ligand-receptor interaction may also contribute to transendothelial migration. Homophilic interactions have been also reported to facilitate extravasation. Jouve et al. showed that expression of CD146 in melanoma and endothelial cells supports metastasis into the lungs, through increased production of VEGF-2 [44]. Hemodynamic shear stress increases the production of reactive oxidation species and extracellular signal-regulated kinases that promote the migration of tumor cells [45].

### 2.6. Colonization

After detachment from the primary tumor, metastatic cancer cells infiltrate and colonize different organs. The gap between infiltration and colonization is latency. The prolonged period of latency implies greater malignant evolution of disseminated tumor cells and/or their microenvironment before colonization [1]. When metastasis is fast, like in lung cancer and pancreatic adenocarcinoma, there is a little-to-no capacity for metastatic cells to evolve. Common organs for cancer cell colonization are the liver, brain, bone marrow, and lungs due to circulation patterns [1]. This organotropic feature of cancer cells is favored by the upregulation of cell adhesion molecules such as metadherin that can specifically bind to the pulmonary vasculature to help the CTC to enter the lung tissue [46]. Furthermore, sinusoids in the capillaries of bone marrow have fenestrated endothelia for the passage of blood cells. These structures allow the CTC to enter bone marrow [47]. It has been shown that the transcription factor SNAI2 in glioma gives the tumor cells an ability to metastasize into multi-organs [48]. Once metastatic tumor cells reach the target organ, they undergo a process called mesenchymal-to-endothelial transition (MET) for the localization and proliferation in the metastatic organ. Loss of mesenchymal phenotype gives macrometastatic colonies a capacity to overcome growth arrest during the EMT process. Several genes are known to be involved in the formation of metastatic colonies. For example, an inhibitor of DNA-binding (Id) renders tumor cells self-renewable properties and induces MET and pulmonary colonization in breast cancer cells [49]. Inhibition of the Paired Related Homeobox 1 (Prrx1) gene is also important for tumor cells to obtain stem cell characteristics and metastatic colonization [50].

## 3. The LncRNAs Involved in Metastasis

Figure 2 shows different LncRNA molecules that are implicated in the invasion, migration, and metastasis processes of a variety of cancers.

### 3.1. ANRIL

The *ANRIL* LncRNA was first reported in melanoma with a 403 kb deletion at the *CDKN2A/B* locus (9p21.3) [51]. Due to its important location, near *CDKN2A/B*, it has been numerously studied for the inheritance of several diseases. Tumor suppressor proteins, including p15, p16, and cyclin-dependent kinase, are encoded by *CDKN2A* and *CDKN2B*. These loci are silenced in nearly 40% of human cancers where *ANRIL* mediates oncogenic effects such as cell proliferation, adhesion, and metastasis. Increased expression of *ANRIL* is also associated with chemoresistance [52].

Hua et al. reported that high expression of *ANRIL* in human hepatocellular carcinoma (HCC) tissue is positively associated with histologic grade, cell proliferation, and poor survival rate [53]. Down-regulated *ANRIL* has an opposite effect and increases radiosensitivity and expression of miR-125a in nasopharyngeal carcinoma cells [54]. A recent study has shown that an increased expression of *ANRIL* in multiple myeloma inhibits bortezomib-induced apoptosis via *PTEN* promoter [55]. The lack of data on the role of *ANRIL* in metastatic characteristics of tumors urges more investigations to unravel its effects on this cancer hallmark.

### 3.2. CASC2

The LncRNA cancer susceptibility candidate 2 (*CASC2*) is a novel tumor suppressor with the ability to hamper invasion, migration, and metastasis in HCC cells by suppressing EMT [56]. In pancreatic cancer, *CASC2* is involved in the suppression of invasion and metastasis by upregulating *PTEN* and downregulating *miR-21* [57]. In breast cancer, *CASC2* suppresses cell proliferation and metastasis by targeting two different mechanisms, which involve the TGF-β signaling and *miR-96-5p*/*synoviolin* pathways [58,59]. Under-expression of *CASC2* is correlated with the serous histological subtype, lymph node metastasis, poor histological grade, and large tumor size in ovarian cancer samples [60]. The LncRNA *CASC2* acts as a tumor suppressor in esophageal squamous cell carcinoma by inhibiting proliferation, migration, and invasion in these cancer cells [61]. The role of *CASC2* as a tumor suppressor has also been established in several cancer types, including thyroid cancer, lung cancer, bladder cancer, osteosarcoma, and oral squamous cell carcinoma by suppressing the proliferation and metastasis [62,63,64,65].

### 3.3. GAS5

Growth arrest-specific 5 (*GAS5*) is a LncRNA with the ability to induce cell death by binding glucocorticoid receptors. In colorectal cancer cells, it binds *YAP* and *YTH* N6-Methyladenosine RNA Binding Protein 3 (*YTHDF3*) to inhibit cancer progression [66]. The *GAS5* promotes apoptosis in triple-negative breast cancer, which is highly metastatic breast cancer, by binding *miR-378a* [67]. This LncRNA also has anti-invasive effects on ovarian cancer by suppressing *miR-96-5p* and promoting the PTEN/mTOR signaling pathway [68]. In melanoma cancer cells, *GAS5* inhibits metastasis by reducing the expression of MMP-7 and 9, which are two important markers of cancer metastasis [69]. It also prohibits EMT in osteosarcoma cancer [70]. In a recent study, Xu et al. reported that reduced expression levels of GAS5 in papillary thyroid carcinoma decreased tumor cell growth, migration, and lymph node metastasis of cancer cells via the IFNγ/STAT1 signaling pathway [71].

### 3.4. HOTAIR

The HOX antisense intergenic RNA (*HOTAIR*) is a LncRNA that is transcribed from the antisense strand of *HOX* gene cluster with the ability to bind the chromatin modification complex. It regulates gene expression in a trans-regulatory fashion. Through enhancer of zeste homolog 2 (*EZH2*), lysine-specific histone demethylase 1A (*LSD1*), and polycomb repressive complex 2 (*PRC2*), *HOTAIR* silences gene expression and histone methylation. Its positive role in the promotion of metastasis, invasion, and tumor cell proliferation by epigenetic regulation of several metastatic genes and protein products has been extensively studied [72,73]. The *HOTAIR* is also upregulated in cancer-associated fibroblast (CAF) due to increased secretion of TGF-β1 [74]. The CAFs play an important role in metastasis, invasion, and drug resistance of different cancers [74].

The *HOTAIR* also suppresses miR-122 expression which instigates activation of cyclin G1 and subsequent cancerous response in HCC [75]. A decrease in the expression of *miR-122* is associated with the progression of HCC by targeting several genes involved in EMT and angiogenesis. These genes include cyclin G1, insulin-like growth factor-1, WNT1, pyruvate kinase M2, and A disintegrin and metalloprotease 10 (ADAM10) [76,77]. Drug resistance in HCC, which is characterized by the overexpression of TGF-β1, p glycoprotein, and breast cancer resistance protein, is associated with upregulation of *HOTAIR*. This is also associated with the promotion of metastasis in HCC cells [78].

A study by Yang et al. showed that *HOTAIR* mediates SNAP23 phosphorylation, activation of mammalian target of rapamycin (mTOR) signaling cascade, and secretion of exosomes in HCC [79]. Exosomes contain various mRNA, miRNA, LncRNA, and some other non-coding RNAs. Tumor cells use exosomes to help the spread and progression of the tumor [79]. The Collagen alpha-1(V) chain gene is upregulated during gastric cancer progression and immune infiltration. These are mediated by *HOATIR* overexpression and subsequent downregulation of miR-1277-5p [80]. Metastasis of squamous cell carcinoma is also supported by the upregulation of *HOTAIR*, which induces tumor invasion and stimulates EMT [81]. A recent study suggested that the knockdown of *HOTAIR* decreases angiogenesis, proliferation, and migration of renal carcinoma. The HOTAIR competitively binds *miR-126* and regulates the expression of epidermal growth factor-like domain multiple 7 (EGFLD7) and metastasis in these cells [82].

### 3.5. HOTTIP

The HOXA Distal Transcript Antisense RNA (*HOTTIP*) is a LncRNA located at the 5′ end of *HOXA* gene cluster with the ability to facilitate the transcription of these genes upon recruitment of WD repeat domain 5/ mixed lineage leukemia (WDR5/MLL). Activation of the *HOXA13* gene promotes tumorigenesis in the tissue by downregulating *miR-30b* [83,84]. Renal cancer is marked with increased expression of *HOTTIP*, which also is an indicator of poor prognoses such as metastasis, increased tumor size, vascular invasion, and reduced overall survival rate [85]. Furthermore, *HOTTIP* upregulates insulin-like growth factor-2 (IGF-2), which has a role in tumor progression [85]. In pancreatic cancer, *HOTTIP* is also upregulated and imposes tumorigenic effects by the promotion of cancer growth, proliferation, migration, and metastasis [86]. The *HOXA9* binds WDR5 and activates the Wnt/β-catenin pathway, which promotes cancer cell progression and the EMT process in pancreatic cancer cells. Stemness of pancreatic cancer cells is regulated by increased expression of *HOTTIP*, as a result of the production of stem cell factors such as NANOG, OCT4, and SOX2 [87]. The *HOTTIP* increases the resistance of pancreatic cancer cells to cisplatin by inhibiting *miR-137*, which increases the resistance of pancreatic cancer cells to cisplatin. Silencing of *HOTTIP* in these cells induces apoptosis and suppresses the growth and metastasis of pancreas tumor [88].

### 3.6. H19

The *H19* LncRNA, located on chromosome 11p15.5, is expressed in fetal and adult periods and is associated with the differentiation of skeletal muscle cells. Its expression is upregulated in hypoxic stress through the p53/HIF1-α signaling pathway. Furthermore, several oncogenes like *ZEB1, HER2, CALN1, MYC,* and *STAT3/EZH2/Catenin* are upregulated with the expression of *H19* LncRNA [89]. It also increases cell viability, motility, growth, migration, invasion, metastasis, EMT, autophagy, cell cycle progression, colony formation, and glucose metabolism [90,91]. It promotes the development of cancer-mediated chronic infection in HCC [92], contributes to EMT in papillary thyroid carcinoma [93], and increases estrogen-mediated cell survival and proliferation in breast cancer [94].

### 3.7. LINC01121

Long intergenic noncoding RNA 01121 (*LINC01121*) is expressed LncRNA with the ability to act as upstream regulator of SIX Homeobox 2 (*SIX2*) gene [95]. The *LINC01121* is substantially overexpressed in breast cancer cell lines compared with healthy breast epithelial cells [96]. Downregulation of *LINC01121* in breast tumors is associated with the inhibition of cell proliferation, cell cycle progression, migration, and invasion in breast cancer cells [96]. High-mobility group protein 2 (HMGA2) is a target gene of miR-150-5p and is significantly overexpressed in breast tumors [97]. This gene encodes a protein with the ability to enhance the proliferation and metastasis of breast cancer cells. The *miR-150-5p* contributes to the suppression of triple-negative breast cancer metastasis through impeding HMGA2 expression [98]. Further studies revealed that *LINC01121* could indirectly upregulate HMGA2 protein expression through the interaction with miR-150-5p [96].

### 3.8. MALAT1

Metastasis associated with lung adenocarcinoma transcript 1 (*MALAT1*), as the name indicates, was primarily known for its role in the survival rate of patients with non-small-cell lung cancer [99]. It is an excellent predictor of tumor invasion and progression [100]. This LncRNA is highly expressed in various cancer types and exerts its tumorigenic effects by blocking the PI3K/Akt pathway and increasing matrix metalloproteinase-9 (MMP-9) [101,102]. Moreover, in neuroblastoma and retinoblastoma cells, upregulation of MALAT1 activates mitogen-activated protein kinase (MAPK) along with the peroxisome proliferator-activated receptor (PPAR), P53-dependent signaling, and the Wnt/β-catenin pathway [103]. This is mediated by increased expression of miR-124 and subsequent activation of the mentioned pathways via slug knockdown [104]. Its expression in tumor tissues is a biomarker for the development and progression of cancer. For instance, it can help to determine the stage and invasiveness of the tumor [105]. It is also positively associated with lung cancer metastasis and resistance to gefitinib and doxorubicin [106]. Xiang et al. showed that the induction of EMT is accompanied by upregulation of TGF-β1 because of *MALAT1* expression in endothelial progenitor cells. The LncRNA MALAT1 regulates TGF-β receptor 2 and the SMAD3 signaling pathway in these cells [107]. On the contrary, Kim and colleagues reported that *MALAT1*, in breast cancer, has a metastasis-suppressing role, which is facilitated by various pro-metastatic transcription factors of transcriptional enhancer associated domain (TEAD) family [108]. Overexpression of TEAD proteins is associated with the activation of several genes responsible for tumor growth and metastasis such as Yes-associated protein 1 (YAP) and Transcriptional co-activator with PDZ-binding motif (TAZ), which are involved in the hippo pathway [109,110], the pathway that is essential for angiogenesis and tissue regeneration.

### 3.9. MEG3

Maternally expressed gene 3 (*MEG3*) is a LncRNA located on chromosome 14q32.3 and is downregulated in human cancers [111]. Wang et al. showed that *MEG3* was remarkably reduced in patients with metastatic papillary thyroid carcinoma. In addition, they revealed that downregulated *MEG3* had a direct correlation with lymph-node metastasis [112]. Jiao et al. revealed that MEG-3 functions as a suppressor of gastric carcinoma cell growth, invasion, and migration. They suggested that *MEG3* suppresses migratory features of gastric cancer cells by modulating EMT in these cancer cells [113]. Overexpression of *MEG3* in human osteosarcoma cell line, MG63, leads to a significant decrease in proliferation and invasion as well as a remarkable increase in apoptosis [114]. The *MEG3* acts as a metastasis suppressor in melanoma by a mechanism that involves miR-21/E-cadherin axis [115]. It is downregulated in melanoma tissues and cell lines and its level is markedly associated with poor prognosis in patients with this disease [116]. In ovarian cancer, *MEG3* prohibits the tumor progression by acting on *miR-30e-3p* and laminin alpha4 [117]. Additionally, *MEG3* suppresses the metastatic progression of several cancers such as breast cancer, lung cancer, and HCC [91,118,119].

### 3.10. NEF

Neighboring enhancer of FOXA2 (*NEF*) is a LncRNA known for its tumor-suppressive role. Several preclinical studies have shown that the upregulation of *NEF* inhibits cancer progression [120,121]. In triple-negative breast cancer, *NEF* is downregulated because of the upregulation of *miR-155* [122]. It is also reported to suppress metastasis in HCC and inhibits cell invasion and migration in osteosarcoma by downregulating *miR-21* [123]. In cervical cancer, the reduced expression of *NEF* is a characteristic of patients with reduced survival rates. Downregulation of *NEF* is associated with increased production of TGF-β1 that induces metastasis of cancer cells [124,125]. This is likely to be achieved by inhibition of the Wnt/β-catenin pathway [126]. Chang et al. reported that serum concentration of *NEF* is negatively correlated with the stage of non-small-cell lung cancer [127].

### 3.11. NKILA

Nuclear Factor-κB Interacting LncRNA (*NKILA*) is an inflammation-induced LncRNA molecule that has been recognized in triple-negative breast cancer cells after exposing them to tumor necrosis factor (TNF)-α and interleukin-1β (IL-1β) [128]. Downregulated expression of *NKILA* is associated with metastasis and invasiveness in breast cancer patients [128]. In HCC, *NKILA* suppresses the metastatic spread of the tumor by inhibiting NF-κB/Slug-mediated EMT in these tumor cells [129]. This LncRNA also hampers the invasion and migration of tongue squamous cell carcinoma cells by blocking the EMT process in these cancer cells [130]. A study on non-small cell lung cancer showed downregulated *NKILA* in tumor samples. The results indicated that *NKILA* suppresses these cells by acting on the NF-κB/Snail signal pathway [131]. The effect of *NKILA* on the inhibition of migration and invasion of malignant melanoma cells was also shown to be achieved by the regulation of the NF-ĸB signaling pathway [132]. Several studies have affirmed the key role of *NKILA* in the invasion, migration, and metastasis of different cancers such as esophageal squamous cell carcinoma, laryngeal cancer, and head and neck cancer [133,134,135].

### 3.12. NRON

The nuclear factor of activated T-cells (NFAT) is a transcription factor present in the extracts of T-cells with a role in cancer invasion, angiogenesis, and differentiation [136,137]. Non-coding repressor of NFAT (*NRON*) is a LncRNA that represses NFAT by inhibiting the transfer of this factor between the nucleus and cytoplasm (nucleocytoplasmic shuttling). By reducing the transfer of NFAT to the nucleus, *NRON* reduces the proliferation and invasion of vascular endothelial cells [138]. The *NRON* levels are markedly increased in some cancers such as bladder cancer, where it facilitates proliferation, migration, differentiation, and metastasis of cancer cells [136]. The LncRNA *NRON* is downregulated in breast tumors compared to healthy tissues [139]. Reduction in *NRON* levels in breast cancer is associated with increased cell invasion and differentiation and reduced apoptosis [140]. Nonetheless, its increased levels had the opposite effect marked by reduced levels of CCND1, CDK4, and Bcl-2 and an increase in Bax and *miR-302b* leading to the inhibition of cancer progression and metastasis [140].

### 3.13. PTTG3P

Pituitary tumor-transforming 3 (*PTTG3P*) is a LncRNA with a confined protein-coding capacity and implications in tumorigenesis of various cancer types. It is noticeably homologous to its parental gene, *PTTG1* [141]. In the resected cervical cancer (CC) tissue, *PTTG3P* and *PTTG1* had considerably higher expression levels in comparison with their paired adjacent healthy counterparts. Furthermore, the invasiveness of CC cells was enhanced by *PTTG3P* through SNAIL upregulation and E-cadherin downregulation [141]. High expression levels of *PTTG3P*, *PTTG1*, and *PTTG2* have been observed in esophageal squamous cell carcinoma (ESCC) patients and cell lines. In addition, there has been a correlation between TNM stage, tumor depth, and lymph node invasion with the elevated expression of *PTTG3P* in ESCC [142]. Interestingly, the results of an in vitro *PTTG3P* gain-of-function study demonstrated that the invasion and migration of ESCC cells were stimulated because of the increased expression of *PTTG3P*. Conclusively, it is suggested that *PTTG3P* functions as an oncogene in ESCC [143]. The considerable up-regulation of *PTTG3P* in colorectal cancer (CRC) tissues is correlated with distant and lymph node metastasis. It has been indicated that the motility of CRC cells might be promoted by *PTTG3P* through downregulation of miR-155-5P [144]. Huang et al. investigated the oncogenic function of PTTG3P in HCC and revealed that *PTTG3P* expression was considerably increased in HCC patients. They showed that *PTTG3P* upregulation was positively associated with TNM stage, tumor size, and poor survival of patients [145]. Further experiments demonstrated that recombinant overexpression of *PTTG3P* leads to enhanced cell proliferation, migration, and invasion in vitro as well as augmented metastasis and tumorigenesis in vivo. Conversely, opposite influences were recorded following the knockdown of *PTTG3P*. Mechanistically, overexpressed *PTTG3P* leads to the activation of PI3K/AKT and its downstream signals, including cell apoptosis, cell cycle progression, EMT markers, and up-regulation of *PTTG1* [145].

### 3.14. SNHG20

Small nucleolar RNA host gene 20 (*SNHG20*) is primarily known for its role in HCC. However, research has shown its role in the pathogenesis of a variety of cancers including bladder, lung, bone, colorectal cancer, and ovarian cancer. For example, the upregulation of SNHG20 is positively correlated with the activation of Wnt/β-catenin signaling, leading to the proliferation and invasion of ovarian cancer cells [146]. This is supported by increased production of cyclin-dependent kinase inhibitor 1 (p21), cyclin D1, N-cadherin, and vimentin. Guo et al. revealed that *SNHG20* downregulates *miR-140-5p* and ADAM10 to activate the MEK/ERK signaling pathway, which leads to subsequent cell proliferation, invasion, and differentiation [147]. In gastric cancer patients, *SNHG20* is correlated with the size of the tumor, lymphatic metastasis, and a lower overall survival rate [148]. Furthermore, overexpression of *SNHG20* activates the PI3K/Akt/mTOR signaling pathway, contributing to tumor progression and stemness in glioblastoma [149]. Several studies have also shown its upregulation in prostate cancer, osteosarcoma, and laryngeal squamous cell carcinoma [150,151,152].

### 3.15. XIST

LncRNA x-inactive specific transcript (*XIST*) inactivates the X chromosome by accumulation near the transcriptional loci of different proteins and contributes to gene silencing [153]. Knockdown of this LncRNA suppresses the growth, proliferation, migration, and invasion of some tumor cells. In glioblastoma stem cells, this effect is mediated by the upregulation of miR-152, indicating a negative correlation between *XIST* and miR-152 [154]. There are several pathways by which *XIST* exerts its effect on different types of cancer such as non-small cell lung cancer, breast cancer, and colorectal cancer [155,156]. In glioma cells, silencing of *XIST* suppresses metastasis and angiogenesis because of increased expression of *miR-429* [157].

A recent study by Xu et al. reported that silencing of *XIST* inhibits lung cancer cell growth by allowing the transcription of p53 and NLR family pyrin domain containing 3 (NLRP3), which suppresses the function of SMAD2 to inhibit its translocation to the nucleus. This LncRNA upregulates the transcription of Bcl2 and reduces that of E-cadherin, which results in the detachment of tumor cells from the primary tissue and its metastasis to distant organs [158]. The LncRNA *XIST* also competes with *miR-744*, which activates the Wnt/β-catenin signaling pathway, leading to tumor progression, invasion, and migration [159].

The LncRNA *XIST* promotes metastasis of breast cancer through different pathways and mechanisms. It inhibits the action of miR-125b leading to an increase in the production of NOD-like receptor family CARD domain containing 5 (NLRC5), which is a known inducer of metastasis in breast cancer [160]. However, Xing et al. showed that loss of *XIST* is associated with metastasis of breast cancer to the brain, via activation of the EMT process and c-Met [161]. Similar findings have been suggested by Zheng et al. [162].

The role of *XIST* in tumor progression and metastasis has also been reported in gastric cancer. It suppresses *miR-101* and modulates the function of EZH2. Additionally, it targets TGF-β1 by repressing *miR-185*, metastasis-associated in colon cancer 1 gene (MACC) via suppression of *miR-497* and JAK expression through competing with *miR-337* [163,164,165,166].

### 3.16. ZFAS1

The ZNFX1 antisense RNA 1 (*ZFAS1*) is a novel LncRNA transcribed in the antisense orientation of zinc finger NFX1-type containing 1(ZNFX1). The LncRNA *ZFAS1* is upregulated in several cancers and may contribute to the development and progression of these cancers [167]. In prostate cancer, knocking down *ZFAS1* suppresses the migration and invasion of these cancer cells by inhibiting EMT [168]. Upregulation of *ZFAS1* induces colorectal cancer cell migration, invasion, and metastasis and is positively correlated with TNM stage these tumors [169]. The *ZFAS1* also stimulates proliferation and metastasis in pancreatic cancer cells by acting on *miR-497-5p* [170]. In colorectal cancer, the upregulated level of *ZFAS1* is directly associated with poor prognosis and promotes invasion and metastasis [169]. The *ZFAS1* is involved in colorectal cancer progression by inducing vascular endothelial growth factor A (VEGFA), which is one of the important inducers of angiogenesis in tumors [171]. Moreover, *ZFAS1* acts as a tumor suppressor in breast cancer and its downregulated level is associated with augmented proliferation and metastatic breast tumors [172]. Some data affirm that *ZFAS1* is a major modulator of the EMT process in colon adenocarcinoma [173].

## 4. Phytochemicals That Target LncRNAs to Cease Metastasis

During the last decade, numerous bioactive compounds have been studied for their potential activities against metastasis through modulating LncRNAs (Table 1, Figure 3). Some of the more frequent cancer models in which these phytochemicals have been examined to regulate metastasis by acting on LncRNAs include breast cancer, hepatocellular carcinoma, and prostate cancer [174].

Betulinic acid (BA) is a natural component derived from the outer bark of a variety of tree species like white-barked birch [186]. A recent in vivo and in vitro study conducted to evaluate the effects of BA on HCC cells revealed that this triterpenoid can suppress the progression and invasion of these cells through inhibition of *MALAT1* expression. These effects were shown to be exerted in a dose-dependent manner. Moreover, Hematoxylin and eosin staining and immunohistochemical (IHC) observations were applied to visualize tumor tissues in a BALB/c nude mice model of HCC. In the mice treated with BA, a well-defined layer of tumor tissues without notable invasion was recognized, while the untreated group showed observable invasion in their tumor tissues. This was suggestive of the inhibitory role of BA on HCC cell invasion. Likewise, Ki67 expression, as a cell proliferation marker, was suppressed in BA-treated mice compared with untreated animals [187]. The biological activities of BA warrant more studies to prove an eventual translation into clinical settings, which can lead to the identification of a novel therapeutic approach for cancer patients [188]. Its potential anti-cancer effects led to the conduction of a phase I/II clinical trial to investigate the effect of 20% BA ointment on the treatment of moderate to severe forms of dysplastic nevi with no reported results. However, its challenging extraction process and poor water solubility limit its potential application as an anti-cancer drug [175].

Bharangin is a diterpenoid quinone methide, which is derived from the roots of the *Pygmacopremna herbacea* plant [189]. The results of an investigation revealed that MDA-MB-231 cells treated with bharangin could reduce the migration capacity of these cells compared with non-treated cells. The expression of tumor suppressor LncRNAs including *GAS-5* and *MEG3* was significantly augmented in bharangin-treated cells, while this diterpenoid down-regulated oncogenic *H19* LncRNA. They also reported that bharangin had a remarkable potential to inhibit the activation of okadaic acid-induced NF-κB in breast cancer cells [190].

Curcumin (diferuloylmethane) is an ingredient in yellow spice turmeric (*Curcuma longa*) [191]. This polyphenol has different biological activities against numerous human diseases, including cancer. In a recent study, curcumin effectively reduced *H19*-induced EMT in MCF-7/TAMR cells by downregulating N-cadherin and upregulation of E-cadherin, which are two well-known EMT biomarkers. Moreover, wound healing and transwell assays demonstrated that curcumin considerably reduces the migration and invasion of these breast cancer cells [176]. Accordingly, available data from an in vitro study support the key role of curcumin in the suppression of *HOTAIR*-induced migration of renal cell carcinoma (RCC) cell lines [192]. Two RCC cell lines were utilized including 769-P-*HOTAIR* and 769-P-vector cells with high and stable *HOTAIR* expression. The migration capacity of 769-P-HOTAIR cells was substantially higher than that of 769-P-vector cells. Interestingly, curcumin prohibited the migration of 769-P-HOTAIR cells in a concentration-dependent manner [192]. Zamani and colleagues studied the effect of dendrosomal curcumin (DNC) on *MEG3* expression in HCC cells. They observed that DNC effectively augments the expression levels of *MEG3* via upregulation of mir-29a and mir-185 [177]. However, more investigations are needed to confirm the effect of DNC-induced *MEG3* upregulation in the suppression of metastasis in this cancer.

Several clinical trials have been performed to assess the effectiveness of curcumin in treating different cancer types. According to the literature, the nitric oxide (NO) level is associated with different stages of malignancies and increased levels of NO have been reported in leukemic patients [178]. Therefore, Ghalaut et al. conducted a clinical study to assess the effectiveness of imatinib alone or in combination with turmeric powder on the levels of NO in 50 patients with chronic myeloid leukemia (CML). Twenty-five patients were treated with imatinib (400 mg twice a day) alone, and 25 subjects received imatinib in combination with turmeric powder (5 g three times/day) for 42 days. A more significant decrease in the serum levels of NO in the group with a combined treatment suggested that turmeric powder can be used as an adjuvant in reducing NO levels and may be effective in the treatment of CML [179].

Another randomized, double-blind placebo-controlled clinical trial evaluated the effect of curcumin (4 g daily) on free light-chain ratio response and bone turnover in patients with monoclonal gammopathy of undetermined significance (MGUS) and smoldering multiple myeloma (SMM). The data showed that curcumin could reduce disease procession in these patients [193]. Mahammedi et al. conducted a Phase II trial for examining the efficacy of docetaxel/prednisone for six cycles in combination with curcumin (6 mg per day) in 30 patients with metastatic prostate cancer. Their observations revealed a prostate-specific antigen (PSA) response in 59% of patients and significant efficacy of curcumin in treating cancer with a high response rate, well tolerability, and patient acceptability [194]. The safety profile and tolerability of curcumin was explored in a Phase I/II trial in metastatic colorectal cancer patients. The results showed that oral curcumin (2 g daily) with 12 cycles of 5-fluorouracil, folinic acid, and oxaliplatin chemotherapy regimen is safe and tolerable [195].

In another Phase II open-label clinical trial, the immunomodulatory efficacy of 100 mg of curcuminoids (extracted from *Curcuma longa* root) was assessed for tumor-induced inflammation in seven patients with endometrial carcinoma. The levels of inflammatory biomarkers in the patients who received this regimen were significantly suppressed and this may indicate curcumin-based compounds as supplementary regimens in endometrial carcinoma [196]. A Phase 1 clinical study on the chemopreventive potential of curcumin (4 g daily for 4 weeks) in colorectal cancer was conducted in 2010. Forty patients were enrolled to be participated in this study. No results have yet been reported for this study [197]. A Phase I study aimed to evaluate the short-term effects of supplementation with a turmeric extract, Curcumin C3 Complex^®^, on the biomarkers of head and neck squamous cell carcinoma (HNSCC). The tumor samples’ adjacent tissues were used to measure the concentrations of curcumin and its metabolites in patients. The results revealed that this curcumin derivative could be used as a cancer preventing agent in smokers and tobacco users who are at risk of oral cancer [198]. A Phase II trial on effect of a curcumin derivative (Meriva^®^, 500 mg twice daily) was conducted for chemotherapy-treated breast cancer patients undergoing radiotherapy. The activity of nuclear factor-κB and its downstream modulators were quantified after treatment of patients with curcumin. No final results have yet been released [199].

Genistein, as a flavonoid compound derived from soybeans, targets an oncogenic LncRNA *HOTAIR* to suppress the migration and invasion of prostate tumor cells [200]. The tumor suppressor miR-34a, which binds to the *HOTAIR* mRNA sequence, participates in the anti-metastatic mechanism of genistein. Notably, genistein increases the expression of *miR-34a*, which in turn downregulates *HOTAIR* expression and thus suppresses the cell movement capacity of prostate cancer cells [201]. As reported in the literature, genistein changes the levels of phosphorylated tyrosine residues in cellular proteins. Accordingly, a Phase I clinical trial was conducted to determine the pharmacokinetic of two isoflavone preparations, PTI G-2535 and PTI G-4660 (which contained 43% and 90% genistein, respectively), in 13 patients with metastatic prostate cancer. The study also evaluated the toxicity and levels of protein-tyrosine phosphorylation in peripheral blood samples of the patients. Moreover, cohorts of four patients were administered genistein at three doses of 2, 4, or 8 mg/kg daily. The toxicity test results showed that one case with a treatment-related rash. Besides, a significant increase in tyrosine was identified in blood samples of the patients. This may suggest a potential anti-metastatic activity for genistein [202]. In another clinical study, Miltyk et al. investigated the probable genotoxic effect of a purified soy unconjugated isoflavone mixture containing in genistein, daidzein, and glycitein on 20 men with prostate cancer. The patients received 300 mg genistein for 28 days and then with 600 mg/d for another 56 days. Fluorescence in situ hybridization technique was used to measure genotoxicity markers in peripheral lymphocytes. Based on their data, no remarkable toxic changes were observed in genistein-treated patients. Therefore, the authors reported no toxic effects for the mentioned isoflavone mixture despite the in vitro genotoxicity that has been reported in the literature [203].

In a recent study, the impact of IDET, a sesquiterpene lactone extracted from *Elephantopus scaber* [180], on breast cancer cell migration was examined [204]. Data reported from scratch (wound healing) assay showed that IDET could significantly prevent the invasiveness of MDA-MB-231 cells. The healthy breast epithelia abundantly express the LncRNA *NKILA* and long GAS5, but their low expression correlates with metastasis of breast cancer [128,205]. Interestingly, IDET administration remarkably enhanced the expression levels of these LncRNAs [204]. Furthermore, the expression of oncogenic LncRNA *H19*, which is constitutively expressed in various tumor types like breast cancer, was significantly down-regulated due to the treatment of MDA-MB-231 cells with IDET [204]. This suggests that the up-regulation of tumor suppressor LncRNAs and down-regulation of oncogenic LncRNAs by IDET may contribute to motility suppression of MDA-MB-231cells. Likewise, elevated expression levels of oncogenic LncRNAs such as *ANRIL* and *HOTAIR* were observed in the serum samples and clinical tumor tissues of breast cancer patients compared to their paired healthy controls [206]. Nevertheless, the results of a study by Verma and colleagues showed the increased expression of these oncogenic LncRNAs by IDET. This may indicate a compensatory mechanism in response to the suppressed expression of other oncogenic LncRNAs and upregulation of tumor suppressor LncRNAs [204].

As described before, EMT is considered a vital stage in the metastatic propagation of all cancer types. Huang et al. showed that treatment of MCF7 cells with a phytochemical compound (extracted from different plants including grapes, blueberries, and peanuts) known as Pterostilbene [181] impedes EMT through downregulation of *HOTAIR*, *LINC01121*, and *PTTG3P*, as well as upregulation of *MEG3* [207].

A polyphenolic phytoalexin, known as resveratrol, is extracted from a variety of herbs, including berries, grapes, peanuts, pistachio, plums, and white hellebore [208]. In an investigation conducted by Ji et al., in situ hybridization confirmed that there are significantly higher *MALAT1* expression levels in tumor tissues compared to adjacent normal tissues. Besides, they found a statistically significant correlation between the extent of tumor metastasis and invasion with *MALAT1* expression. They also demonstrated that resveratrol can remarkably suppress migration and invasion of human colon cancer cell line LoVo through *MALAT1*-mediated Wnt/β-catenin signaling and its downstream targets in a dose-dependent manner. Overexpression of *MALAT1* using recombinant lentiviral-based experiment confirmed that this oncogenic LncRNA impedes the inhibitory impact of resveratrol on migration and invasion of LoVo cells [182]. Several clinical trials have been done to explore the effects of resveratrol or resveratrol-reached plant extracts on different types of cancer. A phase I trial evaluated the safety, tolerability, and dose determination of muscadine grape skin extract, which contains resveratrol, in men with recurrent prostate cancer (BRPC). Of 14 patients, seven remained in the study and received 4000 mg of the extract. According to the results, the extract led to a delayed disease recurrence by lengthening the PSA doubling time by 5.3 months. The safety assessments showed four patients with gastrointestinal symptoms, including grade 1 flatulence, grade 1 soft stool, and grade 1 eructation [209]. Nguyen et al. designed a clinical trial to study the effects of freeze-dried grape powder (GP) (containing resveratrol and resveratrol derived from plants) on the expression of factors involved in the Wnt pathway in 8 colorectal cancer patients. Treatment of the patients with GP (80 g/day containing 0.07 mg of resveratrol) for 14 days downregulated the expression of the Wnt target genes within regular mucosa of the patients’ samples. According to the results, the authors suggested GP or resveratrol as colon cancer preventing compounds [183]. In another study, Patel et al. treated 20 colorectal cancer patients with resveratrol at 0.5 or 1 g/day for eight days. Then they quantified the expression of proliferation marker Ki-67 in tumor tissues. The results showed a 5% decline in the expression levels of Ki-67 in tumor tissues, indicating tumor suppressing activity of resveratrol in colorectal cancer patients [210]. Howells et al., in a Phase I randomized, double-blind pilot clinical trial, studied the effect of a resveratrol derivative (SRT501) on colorectal cancer patients with hepatic metastases. They clarified that SRT501 at a dose of 5 g/day for two weeks upregulated caspase-3 within liver tissue. This may suggest a pro-apoptotic activity for this resveratrol derivative in this cancer type [211]. Popat et al. conducted a Phase II trial to assess the possible activity of another resveratrol (SRT501) in combination with bortezomib in patients with relapsed and or refractory multiple myeloma. This resveratrol formulation was administered to 24 participants at a dose of 5 g/day for 20 days in a 21-day cycle up to 12 cycles. The results of the study indicated an unacceptable safety profile and minimal efficacy in these patients [212]. In a randomized placebo controlled clinical trial, Kjaer et al. treated 66 patients with prostate hyperplasia with two doses of resveratrol (150 or 1000 mg/day) for 4 months. Their data revealed that resveratrol treatment significantly decreased serum levels of androstenedione, dehydroepiandrosterone, and dehydroepiandrosterone-sulphate, but no remarkable effect was observed in prostate sizes [213].

Sanguinarine is an alkaloid derived from Bloodroot (*Sanguinaria canadensis*) with a significant inhibitory activity against the migratory ability of ovarian epithelial cancer cells [214]. In an in vitro experiment conducted by Zhang et al. to evaluate the potential effects of sanguinarine on human ovarian SKOV3 cells, this alkaloid inhibited the viability, migration, and invasion of these cells and increased apoptosis as well. Interestingly, *CASC2* is induced by this alkaloid and silencing *CASC2* rescues the antitumor effects of sanguinarine. This process was suggested to be mediated through *CASC2*–EIF4A3 signaling and/or PI3K/AKT/mTOR and NF-κB signal transductions [215]. 

Silibinin, a bioactive component isolated from the seeds of milk thistle (*Silybum marianum*), also has inhibitory potential against bladder cancer [216]. The *HOTAIR* expression is augmented by KRAS [184] and the PI3K pathways [217], and silibinin imposes its inhibitory effects on *HOTAIR* and *ZFAS1* by decreasing the activity of actin cytoskeleton and PI3K/Akt signal transductions in bladder cancer cells [218]. In a clinical study, Barrera et al. treated two patients with brain metastases from non-small cell lung cancer (NSCLC) with silibinin-based nutraceutical (Legasil). They found that Legasil treatment could significantly improve the clinical and radiological data of these patients. They also observed that silibinin treatment of the patients not only suppressed progressive brain metastases and reduced peritumoral brain edema but also did not affect the size of NSCLC tumors. The authors suggested that the combination of brain radiotherapy and Legasil may be a promising regimen to reduce brain edema and can provide local control and time for seeking other potential therapies for these patients [219]. Siegel et al. conducted a phase I study of silibinin phosphatidylcholine to determine the maximum tolerated dose per day of the compound in patients with advanced HCC. The serum levels of silibinin and silibinin glucuronide were increased within 1 to 3 weeks but all three patients died within 23–69 days of enrolling into the trial and no remarkable data were found in this study [185]. Flaig et al. enrolled 12 patients with prostate cancer to a trial study to estimate the tissue and blood effects of high-dose silibinin-phytosome in prostate cancer. Six patients were treated with silibinin at a single dose of 13 g/day for 20 days and six additional subjects were served as a control. The results revealed that high-dose silibinin led to high blood levels transiently, but low concentrations of the compound were observed in prostate tissue, indicating a weak penetration of silibinin into the prostate tissue [220].

## 5. Conclusions

Recent data shows that epigenetic role-players such as LncRNAs play key roles in regulating the malignant transformation and progression of cancers. Given that LncRNAs have pivotal roles in the modulation of a variety of cellular processes, more explorations are needed to unravel their possible mechanism of action in these processes. As has been numerously reported in the literature, phytochemical compounds from natural plants show potential effects on LncRNAs. Phytochemical compounds have been thus demonstrated to modulate the balance of expression of both oncogenic and antitumor LncRNAs, resulting in an anti-metastatic and anticancer effect (Figure 4). However, it appears that direct cellular target molecules of the phytochemicals and their exact mechanism of action are not known. Although well documented, the anti-metastatic effect of phytochemical compounds demands more preclinical and clinical studies to confirm their potential and further identify their molecular mechanism(s) of action. The evidence reviewed herein implies that targeted therapies using cancer-related LncRNAs could lead to the development of novel and effective treatment strategies for different types of cancer. Due to the important roles of LncRNAs in different cellular processes, phytochemicals that target these molecules may also boost the sensitivity of tumors to therapeutic methods.

## Figures and Tables

**Figure 1 molecules-28-00987-f001:**
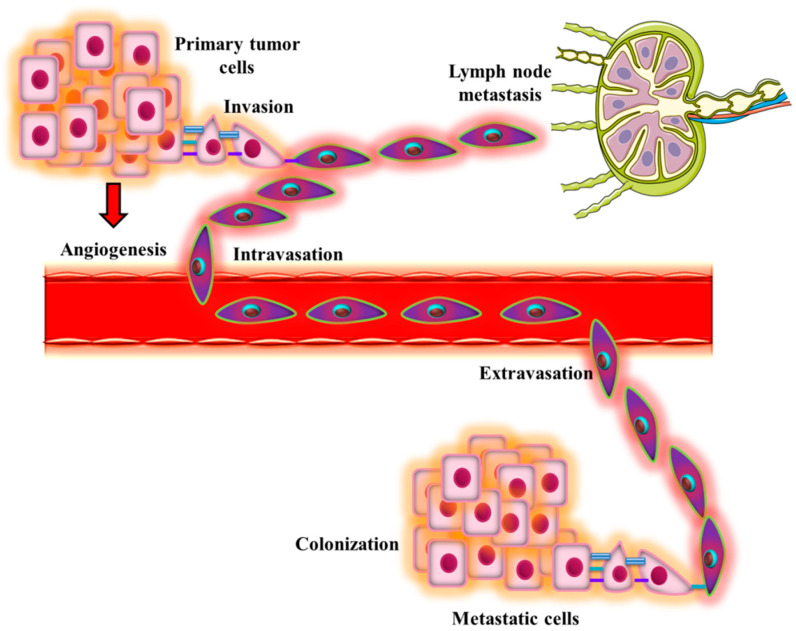
Different steps of cancer cell metastasis. Primary tumor cells invade the adjacent tissue to migrate away from the primary site. Afterward, they intravasate into blood vessels or enter lymph nodes and get transported to the other organs. Circulating tumor cells then extravasate into the secondary organ and start to proliferate to form a colony in the metastatic site.

**Figure 2 molecules-28-00987-f002:**
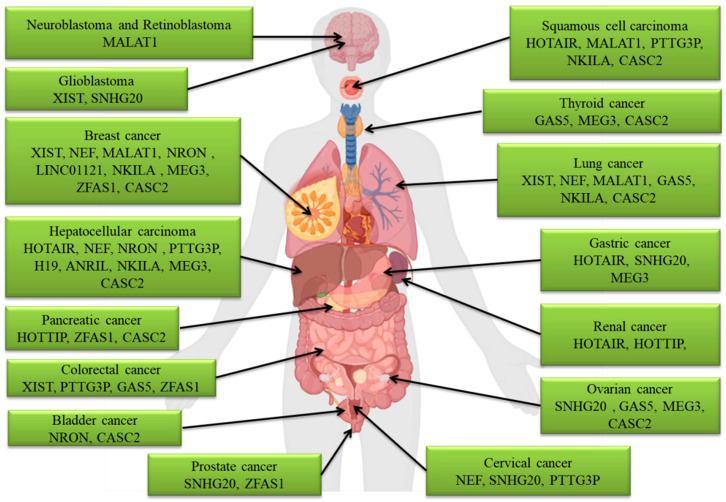
LncRNAs with a role in the invasion, migration, and metastasis of different types of cancers.

**Figure 3 molecules-28-00987-f003:**
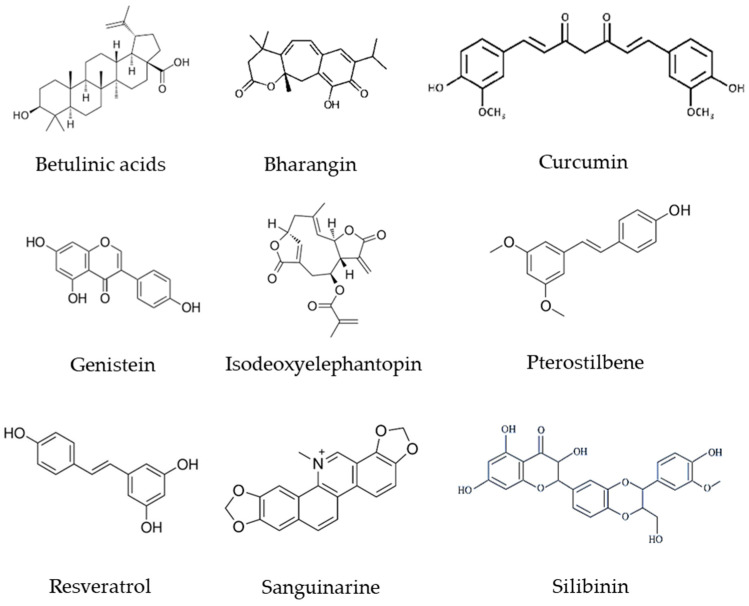
Chemical structure of phytochemicals involved in the invasion, migration, and metastasis of various cancers.

**Figure 4 molecules-28-00987-f004:**
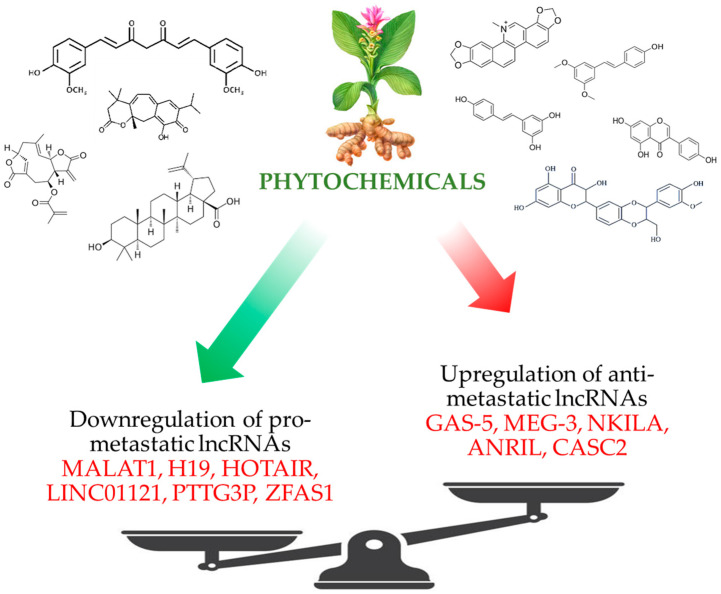
Schematic representation of the effects of phytochemicals on the balance between the expressions of pro- and anti-metastatic LncRNAs.

**Table 1 molecules-28-00987-t001:** Phytochemicals targeting LncRNAs to inhibit cancer metastasis.

Phytochemicals	Plant Source	Concentration	LncRNAs	Alteration	Cancer Type	Cancer Model	Ref.
Betulinic acids	Outer bark of a variety of tree species like white-barked birch	PLC/PRF/5 cell line: IC_50_ = 63.04 μM for 48 h MHCC97L cell line: IC_50_ = 40.02 for 48 h Mice: 10 mg/kg/day	*MALAT1*	Down-regulated	Hepatocellular Carcinoma	BALB/c nude mice, PLC/PRF/5 and MHCC97L cell lines	[175]
Bharangin	*Pygmacopremna herbacea*	0, 1, 2.5, and 5 μM for 24 h	*GAS-5* *MEG3* *H19*	Up-regulated Up-regulated Down-regulated	Breast Cancer	MCF-7 cell line	[176]
Curcumin	*Curcuma longa* (turmeric)	0, 5, 15, and 20 μM for 48 h	*H19*	Down-regulated	Breast Cancer	MCF-7/TAMR * cell line	[177]
Curcumin	*Curcuma longa* (turmeric)	5 to 10 μM for 24 h	*HOTAIR*	Down-regulated	Renal Cell Carcinoma	769-P-HOTAIR and 786-0 cell lines	[178]
DNC *	*Curcuma longa* (turmeric)	0–25 μM for 48 h	*MEG3* *HOTAIR*	Up-regulated Down-regulated	Hepatocellular Cancer	HuH-7 and HepG2 cell lines HuH-7 cell line	[179]
Genistein	Soybean	25 µM for 48 h	*HOTAIR*	Down-regulated	Prostate Cancer	PC3, DU145 cell lines	[180]
IDET *	*Elephantopus scaber* Linn	1, 2.5 and 5 µM for 24 h	*NKILA* *GAS-5* *H19* *HOTAIR* *ANRIL*	Up-regulated Up-regulated Down-regulated Up-regulated Up-regulated	Breast Cancer	MDA-MB-231 cell line	[129,181]
Pterostilbene	Grapes, blueberries, and peanuts	0, 1, 5, 25, and 50 μM for 24 h	*MEG3* *HOTAIR* *LINC01121* *PTTG3P*	Up-regulated Down-regulated Down-regulated Down-regulated	Breast Cancer	MCF7 cell line	[182]
Resveratrol	Berries, grapes, peanuts, pistachio, plums, and white hellebore	IC_50_ = 55 µM for 24 h	*MALAT1*	Down-regulated	Colorectal Cancer Cells	LoVo cell line	[183]
Sanguinarine	*Sanguinaria canadensis* (Bloodroot)	0–5 µM for 24 h	*CASC2*	Up-regulated	Epithelial Ovarian Cancer	SKOV3 cell line	[184]
Silibinin	*Silybum marianum* (Seeds of milk thistle)	10 µM for 24 h	*HOTAIR* *ZFAS1*	Down-regulated	Bladder Cancer	T24, UM-UC-3 cell lines	[185]

* IDET: Isodeoxyelephantopin, MCF-7/TAMR: MCF 7/tamoxifen-resistant cell, DNC: Dendrosomal curcumin (Nanocurcumin).

## Data Availability

Not applicable.

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
