# Peer review of "Long Non-Coding RNAs as Novel Targets for Phytochemicals to Cease Cancer Metastasis"

_molecules, 2023, doi:10.3390/molecules28030987_

Round 1

Reviewer 1 Report

Major revision required   There is much written about the metastasis however, data regarding the mentioned topic is too less.    Review will be made too lengthy by adding majority of the metastasis data which is already mentioned in previous reviews.   1. What is the main question addressed by the research?
    2. Do you consider the topic original or relevant in the field? Does it address a specific gap in the field? The theme is OK but the details of of main topic is too less   3. What does it add to the subject area compared with other published material? Its add the value to long non coding RNA to inhibit cancer metastasis, but majorit of the part seems un appropriate   4. What specific improvements should the authors consider regarding the methodology? What further controls should be considered? Focus more on long non coding RNA therapeutics them detail about its involvement in metastasis      5. Are the conclusions consistent with the evidence and arguments presented and do they address the main question posed? Yes   6. Are the references appropriate?
Yes   7. Please include any additional comments on the tables and figures.

Tables need to be updated and add more reference to highlight more on the therapeutic potential to control Lnc RNA

Author Response

Dear Editor, Dear Reviewers,

  On behalf of all coauthors and co-corresponding authors, I am submitting the enclosed revised manuscript entitled: Long non-coding RNAs as novel targets for phytochemicals to cease cancer metastasis

for consideration as an Review for publication in Molecules.

We sincerely thank the reviewer for constructive criticisms and valuable comments, which were of great help in revising the manuscript. Accordingly, the revised manuscript has been systematically improved with new information and additional interpretations. Our responses (AC) to the reviewer’s comments (RC) are given below.

We hope the review will be found suitable for publication in Biomolecules.

.

Sincerely yours,
Dr Marc Maresca

Reviewer 1

(RC) There is much written about the metastasis however, data regarding the mentioned topic is too less.  Review will be made too lengthy by adding majority of the metastasis data which is already mentioned in previous reviews.

(AC) With all due respect to the reviewer, we would like to explain that describing the metastasis process in the present review article will help the readers to understand the relationship between lncRNAs and metastasis and the clear-cut effect of phytochemicals on this process by regulating lncRNAs. We explained the metastasis process in this present manuscript in detail to give the readers a clear view of the involvement of the mentioned lncRNAs in different steps of the metastasis process. Then, we introduced the phytochemicals that target these molecular targets to affect various stages of metastasis. This can help the readers to clearly understand the central concept of this review article in a step-by-step way. Besides, we have not listed all of lncRNAs which have been reported in the literature to be contributed to cancer metastasis but rather have listed those lncRNAs that have been reported as targets for phytochemicals to inhibit metastatic cascade. However, in accordance with the reviewer’s suggestion, we have added more information to the third part of the manuscript to improve and highlight the role of phytochemicals.

(RC) 1. What is the main question addressed by the research?

(AC) Are there any potential phytochemical with ability to tackle cancer metastasis by targeting lncRNAs, as one of the main regulators of epigenetic alterations in cancer?

(RC) 2. Do you consider the topic original or relevant in the field? Does it address a specific gap in the field? The theme is OK but the details of main topic is too less.

(AC) We have answered this question in the first comment.

(RC) 3. What does it add to the subject area compared with other published material? Its add the value to long non coding RNA to inhibit cancer metastasis, but majorit of the part seems un appropriate

(AC) We have answered this question in the first comment.

(RC) 4. What specific improvements should the authors consider regarding the methodology? What further controls should be considered? Focus more on long non coding RNA therapeutics them detail about its involvement in metastasis  

 (AC) We have answered this question in the first comment.

(RC) 5. Are the conclusions consistent with the evidence and arguments presented and do they address the main question posed? Yes 

(AC) We would like to thank the reviewer for this nice comment.

(RC) 6. Are the references appropriate? Yes  

(AC) We would like to thank the reviewer for this nice comment.

(RC) 7. Tables need to be updated and add more reference to highlight more on the therapeutic potential to control Lnc RNA

(AC) With respect to the reviewer, we should explain that all the available data in the literature regarding the therapeutic potential of phytochemicals by controlling lncRNAs have been explained in this present article and so no more data is available to be added.

Reviewer 2 Report

"Long non-coding RNAs as novel targets for phytochemicals to cease cancer metastasis" by Rajabi et al summarizes many points regarding a potental link between long non-coding RNAs (lncRNAs), phytochemicals, and cancer metastasis. Here are my major and minor points.

Major:

- This manuscript, as a review article, lacks originality which can advance thinking and knowledge. It simply collects unconnected data and information reported elsewhere. For instance, potential links between lncRNAs and metastasis have been reviewed recently and extensively, e.g. 10.1038/s41568-021-00353-1, 10.3389/fonc.2021.641343, 10.3390/ijms22042100, 10.1016/j.bbcan.2021.188519, or 10.3390/ijms21228855. 

- The unique point of this manuscript is the possibility of using phytochemicals as a novel treatments for cancer metastasis via regulating lncRNAs. The text lists several chemicals with their--direct or indirect--effects on regulation of lncRNA genes. Most, if not all, literatures cited in the manuscript were studies conducted in the context of non-cancer and non-metastasis. The authors also fails to describe or propose potential connections or molecular mechanisms for the treatment of cancer metastasis using phytochemicals through lncRNA gene expression control. The authors should discuss a possible crosstalk and potential links between lncRNAs, cancer metastasis, and phytochemicals. Diagrams might help.

Minor points:

- English language editing may be required.

- I found that the authors cited (too?) many references: 211. As mentioned above, this may be because the manuscript was written in a way to list many unconnecteds. If the manuscript is modified as suggested, I believe that the number of references will be much less.

Author Response

Dear Editor, Dear Reviewers,

  On behalf of all coauthors and co-corresponding authors, I am submitting the enclosed revised manuscript entitled: Long non-coding RNAs as novel targets for phytochemicals to cease cancer metastasis

for consideration as an Review for publication in Molecules.

We sincerely thank the reviewer for constructive criticisms and valuable comments, which were of great help in revising the manuscript. Accordingly, the revised manuscript has been systematically improved with new information and additional interpretations. Our responses (AC) to the reviewer’s comments (RC) are given below.

We hope the review will be found suitable for publication in Biomolecules.

.

Sincerely yours,
Dr Marc Maresca

Reviewer2

Major:

(RC) 1. This manuscript, as a review article, lacks originality which can advance thinking and knowledge. It simply collects unconnected data and information reported elsewhere. For instance, potential links between lncRNAs and metastasis have been reviewed recently and extensively, e.g. 10.1038/s41568-021-00353-1, 10.3389/fonc.2021.641343, 10.3390/ijms22042100, 10.1016/j.bbcan.2021.188519, or 10.3390/ijms21228855.

(AC) With all due respect to the reviewer, we would like to explain why we do not agree with this perspective. All the papers that have been mentioned by the reviewer have reviewed different roles of lncRNAs in cancer metastasis. However, the present manuscript deals with the potential anti-metastatic properties of natural phytochemicals by targeting lncRNAs, as one of the major epigenetic regulators of cancer metastasis. This is the main novelty or originality of the present review article. We understand the reviewer’s perspective regarding the explanation of the relationship between lncRNAs and cancer metastasis in the present review and this may appear to be repetitive considering other reviews in the literature, but we believe that describing the metastasis process will help the readers to understand the relationship between lncRNAs and metastasis and the clear-cut effect of phytochemicals on this process by regulating lncRNAs. Besides, we have not listed all of lncRNAs which have been reported in the literature to be contributed to cancer metastasis but rather have listed those lncRNAs that have been reported as targets for phytochemicals to inhibit metastatic cascade. We explained the metastasis process in this present manuscript in detail to give the readers a clear view of the involvement of the mentioned lncRNAs in different steps of the metastasis process. Then, we introduced the phytochemicals that target these molecular targets to affect various stages of metastasis. This can help the readers to clearly understand the central concept of this review article in a step-by-step way.

(RC) 2. The unique point of this manuscript is the possibility of using phytochemicals as a novel treatments for cancer metastasis via regulating lncRNAs. The text lists several chemicals with their--direct or indirect--effects on regulation of lncRNA genes. Most, if not all, literatures cited in the manuscript were studies conducted in the context of non-cancer and non-metastasis.

(AC) With respect to the reviewer’s comments, we would like to inform that of 211 references cited in the manuscript; only 13 were in the context of non-cancer and non-metastasis. Thus, most of the references are clearly about the cancer and metastasis.

(RC) 3. The authors also fails to describe or propose potential connections or molecular mechanisms for the treatment of cancer metastasis using phytochemicals through lncRNA gene expression control. The authors should discuss a possible crosstalk and potential links between lncRNAs, cancer metastasis, and phytochemicals. Diagrams might help.

(AC) Based on the reviewer’s request, we carefully looked at the manuscript to give potential connections and/or molecular mechanisms related to the phytochemicals and metastatic cascades used by lncRNAs. However, we found that the interconnecting factors are missing from most of the studies reviewed in this paper. Therefore, it appears that designing a diagram or schematic illustrations of these interconnections or cornstalks between the above-mentioned parameters is not feasible here. However, we are eager to have additional valuable comments from the reviewer’s side to improve the manuscript.

Minor points:

(RC) 4. English language editing may be required.

(AC) In accordance with the reviewer’s suggestion, we have revised the grammatical errors of the whole manuscript.

(RC) 5. I found that the authors cited (too?) many references: 211. As mentioned above, this may be because the manuscript was written in a way to list many unconnecteds. If the manuscript is modified as suggested, I believe that the number of references will be much less.

(AC) We understand the concerns of the reviewer, but as suggested by other reviewers, we had to add some paragraphs to the manuscript and this may increase the number of references inevitably.

Reviewer 3 Report

A review report of “Long non-coding RNAs as novel targets for phytochemicals to cease cancer metastasis” by Rajabi et al

Rajabi et al. review metastasis steps and lncRNAs involved in metastasis followed by phytochemicals that affect expression of the lncRNAs. Since it appears that direct cellular target molecules of the phytochemicals and their action mechanism are not known, it would be better to discuss this point. This paper would become more valuable by describing current clinical studies for the phytochemicals.

Comments

1. Journal names of many references are missing.

2. It may be better to replace some references by the latest ones (e.g. ref. 1, 2, 25-30).

3. Line 91: “extravasation” should be “intravasation”.

4. Figure 1 should be moved to after line 77.

5. Lines 204 and 251: The definition of “HCC” should be in line 204.

6. Figure 4: The balance in the schematic may be misleading because it looks like differences in absolute amounts of different lncRNAs are shown.

Author Response

 Dr Marc Maresca

Tél : +33(0) 4 91 28 91 96

Courriel : [email protected]

Marseille, the 31th December 2022,

Dear Editor, Dear Reviewers,

  On behalf of all coauthors and co-corresponding authors, I am submitting the enclosed revised manuscript entitled: Long non-coding RNAs as novel targets for phytochemicals to cease cancer metastasis

for consideration as an Review for publication in Molecules.

We sincerely thank the reviewer for constructive criticisms and valuable comments, which were of great help in revising the manuscript. Accordingly, the revised manuscript has been systematically improved with new information and additional interpretations. Our responses (AC) to the reviewer’s comments (RC) are given below.

We hope the review will be found suitable for publication in Molecules.

.

Sincerely yours,
Dr Marc Maresca

Reviewer3

Rajabi et al. review metastasis steps and lncRNAs involved in metastasis followed by phytochemicals that affect expression of the lncRNAs.

(RC) 1. Since it appears that direct cellular target molecules of the phytochemicals and their action mechanism are not known, it would be better to discuss this point.

(AC) We appreciate the reviewer’s perspective. So, we have added a brief description to illustrate this point in the conclusion section.

(RC) 2. This paper would become more valuable by describing current clinical studies for the phytochemicals.

(AC) In accordance with the reviewer’s suggestion, we have added some available clinical data to the manuscript.

(RC) 3. Journal names of many references are missing.

(AC) We would like to thank the reviewer for pointing out this error. We have corrected all references.

(RC) 4. It may be better to replace some references by the latest ones (e.g. ref. 1, 2, 25-30).

(AC) In accordance with the reviewer’s suggestion, we have replaced the mentioned references by new ones.

(RC) 5. Line 91: “extravasation” should be “intravasation”.

(AC) In accordance with the reviewer’s suggestion, we have revised it.

 (RC) 6. Figure 1 should be moved to after line 77.

(AC) In accordance with the reviewer’s suggestion, we have moved the figure to the requested location.

 (RC) 7. Lines 204 and 251: The definition of “HCC” should be in line 204.

(AC) In accordance with the reviewer’s suggestion, we have revised it.

(RC) 8. Figure 4: The balance in the schematic may be misleading because it looks like differences in absolute amounts of different lncRNAs are shown.

(AC) We understand the perspective of the reviewer. However, the purpose of designing this figure was to illustrate the net effect of phytochemicals on the expression levels of lncRNAs reported in the literature. So as depicted in the figure, all studies that have been done to date show that phytochemicals downregulated more lncRNAs in comparison to upregulated ones.

Round 2

Reviewer 2 Report

Thank you for your the revised manuscript.